# Utility of Presepsin and Interferon-λ3 for Predicting Disease Severity and Clinical Outcomes in COVID-19 Patients

**DOI:** 10.3390/diagnostics13142372

**Published:** 2023-07-14

**Authors:** Gun-Hyuk Lee, Mikyoung Park, Mina Hur, Hanah Kim, Seungho Lee, Hee-Won Moon, Yeo-Min Yun

**Affiliations:** 1Department of Laboratory Medicine, Konkuk University School of Medicine, Konkuk University Medical Center, 120-1, Neungdong-ro, Hwayang-dong, Gwangjin-gu, Seoul 05030, Republic of Korea; leegunhyuk93@gmail.com (G.-H.L.); md.hkim@gmail.com (H.K.); hannasis@hanmail.net (H.-W.M.); ymyun@kuh.ac.kr (Y.-M.Y.); 2Department of Laboratory Medicine, Eunpyeong St. Mary’s Hospital, College of Medicine, The Catholic University of Korea, Seoul 03312, Republic of Korea; mikyoung.pak@gmail.com; 3Department of Preventive Medicine, College of Medicine, Dong-A University, Busan 49201, Republic of Korea; lgydr1@gmail.com

**Keywords:** presepsin, interferon-λ3, COVID-19, severity, in-hospital mortality, SOFA score

## Abstract

We explored the utility of novel biomarkers, presepsin and interferon-λ3 (IFN-λ3), for predicting disease severity and clinical outcomes in hospitalized Coronavirus (COVID-19) patients. In a total of 55 patients (non-critical, *n* = 16; critical, *n* = 39), presepsin and IFN-λ3 were compared with sequential organ failure assessment (SOFA) scores and age. Disease severity and clinical outcomes (in-hospital mortality, intensive care unit admission, ventilator use, and kidney replacement therapy) were analyzed using receiver operating characteristic (ROC) curves. In-hospital mortality was also analyzed using the Kaplan-Meier method with hazard ratios (HR). SOFA scores, age, presepsin, and IFN-λ3 predicted disease severity comparably (area under the curve [AUC], 0.67–0.73). SOFA score and IFN-λ3 predicted clinical outcomes comparably (AUC, 0.68–0.88 and 0.66–0.74, respectively). Presepsin predicted in-hospital mortality (AUC = 0.74). The combination of presepsin and IFN-λ3 showed a higher mortality risk than SOFA score or age (HR [95% confidence interval, CI], 6.7 [1.8–24.1]; 3.6 [1.1–12.1]; 2.8 [0.8–9.6], respectively) and mortality rate further increased when presepsin and IFN-λ3 were added to SOFA scores or age (8.5 [6.8–24.6], 4.2 [0.9–20.6], respectively). In the elderly (≥65 years), in-hospital mortality rate was significantly higher when both presepsin and IFN-λ3 levels increased than when either one or no biomarker level increased (88.9% vs. 14.3%, *p* < 0.001). Presepsin and IFN-λ3 predicted disease severity and clinical outcomes in hospitalized COVID-19 patients. Both biomarkers, whether alone or added to the clinical assessment, could be useful for managing COVID-19 patients, especially the elderly.

## 1. Introduction

Severe acute respiratory syndrome coronavirus 2 (SARS-CoV-2), also known as Coronavirus disease 2019 (COVID-19), outbroke in December 2019. There have been over 765 million cases and 6.9 million deaths worldwide as of 10 May 2023 [1]. Globally, elderly patients aged 65 years or older account for a strikingly higher proportion of COVID-19 deaths (81.0–86.2%), and the mortality rate differed remarkably between the young and the elderly. Compared with patients aged 54 years or younger, the incident rate ratio (IRR) of elderly patients was 62 (95% confidence interval [CI], 59.7–64.7), indicating that the mortality rate of the elderly was 62 times higher than younger patients in cases of COVID-19 [2,3].

COVID-19 patients show heterogenous disease courses ranging from no symptoms to critical illness. The World Health Organization (WHO) stratified COVID-19 severity into four categories: mild, moderate, severe, and critical disease, and sepsis is included in the critical disease category [4]. COVID-19 induces significant changes in the host’s immune response, and sepsis has been observed in nearly all COVID-19 related deaths [5]. Accordingly, the sequential organ failure assessment (SOFA) score that is essential for the diagnosis of sepsis has been also used to predict COVID-19 severity and clinical outcomes [6,7]. COVID-19 patients that have higher SOFA scores and lymphocytopenia upon hospital admission showed a greater risk of developing severe diseases [7]. Along with the SOFA score, several clinical assessment tools have been applied to predict COVID-19 severity and clinical outcomes, including a COVID-19 severity index (CSI), Coronavirus clinical characterization consortium (4C) mortality score, and Veteran’s Health Administration COVID-19 (VACO) index [8,9,10,11]. CSI was developed based on the National Early Warning Score 2 (NEWS-2) for predicting COVID-19 severity in hospitalized patients and showed better capacity for predicting intensive care unit (ICU) admission than NEWS-2 [11].

In addition to the clinical assessments, many conventional and emerging biomarkers have been explored in COVID-19 patients, and biomarkers, alone or added to clinical assessments, are effective options for predicting COVID-19 severity and clinical outcomes [12,13,14,15,16]. Presepsin, a soluble cluster of differentiation 14 subtype (sCD14-ST), is released into circulation by pro-inflammatory signals during infection, and its diagnostic and prognostic utility has been proven in sepsis. Presepsin is also useful for predicting COVID-19 severity and mortality (both in-hospital and 30-day) [13,16,17,18,19]. Interferons (IFNs) are potent multifunctional cytokines secreted by various cell types. COVID-19 blocks type I and III IFN responses by inhibiting innate recognition of the virus, the production of IFNs, and the IFN signaling pathway [20,21,22]. In COVID-19, IFN-λ3 levels surged and dropped suddenly before developing severe disease; IFN-λ3 levels were significantly higher in severe and critical diseases than in mild or moderate diseases and could predict oxygen requirements [23,24].

In this study, we explored the clinical utility of both presepsin and IFN-λ3 for predicting disease severity and adverse clinical outcomes in hospitalized COVID-19 patients. We hypothesized that combined use of these biomarkers or biomarkers added to clinical assessments would be more effective than using the clinical assessment tool alone. We also hypothesized that such an approach might help further stratify disease severity and risk of adverse clinical outcomes even in the same age group, and especially in the elderly.

## 2. Materials and Methods

### 2.1. Study Population

The study population was a total of 55 COVID-19 patients who were admitted to the Konkuk University Medical Center (KUMC), in Seoul, Republic of Korea, from June 2020 to June 2021. They consisted of 30 men and 25 women, and their median age was 72 years (interquartile range [IQR], 61–80 years). The study population had no specific limitations on care at enrollment, and biomarker testing was conducted at enrollment (IQR, 0–1 day). In all patients, remaining samples were collected following routine blood testing, and no additional intervention or blood sampling was performed. Their medical records were reviewed retrospectively for clinical and laboratory findings. The study protocol was approved by the institutional review board of KUMC (approval No. 2021-04-048), and obtaining written informed consent from the study population was waived.

In all patients, the diagnosis of COVID-19 was confirmed via real-time reverse transcription-polymerase chain reaction (RT-PCR) [25]. The Real-Q 2019-nCoV assay (BioSewoom, Seoul, Republic of Korea) with Real-Q 2019-nCoV Detection kits using nasopharyngeal swabs were used, and a positive result was defined as threshold cycle values of ≤38 for the target genes (*E* and *RdRP*), according to the manufacturer’s instructions [25].

Table 1 shows the demographic, clinical, and laboratory findings of the study population. The SOFA scores and CSI were assessed at enrollment together with biomarker testing. Sepsis or septic shock was defined, according to the Sepsis-3 definition [7]. The CSI uses parameters that are scored from 0 to 3 each, and sum of the scored parameters was used to stratify COVID-19 severity (low [0–2], moderate [3–5], high [6–7], and critical [>7]). Parameters used include age, gender, comorbidities (heart failure, chronic obstructive pulmonary disease [COPD], and diabetes with end-organ damage), chest radiography, respiratory rate, pulse oximeter oxygen saturation on room air (SpO_2_ or SpO_2_ in COPD), need of oxygen support, symptoms of dyspnea, D-dimer, lymphocytes, and platelets. Chest radiography was evaluated on admission, but it was reconsidered when a new one was performed. If laboratory test results were available 48 h after admission, they were not considered [12]. According to the CSI, the study population was divided into non-critical (*n* = 16; low, *n* = 3; moderate, *n* = 11; high, *n* = 2) and critical (*n* = 39) diseases.

### 2.2. Presepsin and IFN-λ3 Assays

Residual sera were aliquoted to avoid repeated freezing and thawing and were stored at −70 °C until use. Frozen samples were thawed at room temperature and gently mixed immediately before measuring the presepsin and IFN-λ3 levels. Presepsin and IFN- λ3 levels were measured using the two in vitro diagnostics (IVD) assays, an HISCL Presepsin assay and an HISCL IFN-λ3 assay (Sysmex, Kobe, Japan), with HISCL 5000 automated analyzer (Sysmex), respectively. The HISCL Presepsin assay is based on a delayed one-step sandwich chemiluminescence enzyme immunoassay (CLEIA), and the manufacturer-suggested measurement range was 20 to 30,000 pg/mL [26,27]. The HISCL IFN-λ3 assay is based on two-step sandwich CLEIA, and the manufacturer-suggested measurement range was 3 to 200 pg/mL [23,28]. Both assays were performed according to the manufacturer’s instructions.

### 2.3. Statistical Analysis

Data are presented as a number (percentage) or median (IQR). The Mann–Whitney U test and Fisher’s exact test were used to compare the demographic variables, clinical outcomes, and laboratory data between non-critical (*n* = 16) and critical (*n* = 39) diseases. The distribution of presepsin and IFN-λ3 are presented for the purpose of assessing the normality and detecting outliers using the Shapiro–Wilk test and Grubb’s test. Receiver operating characteristic (ROC) curves were analyzed for the purpose of predicting disease severity and clinical outcomes, and the area under the curve (AUC) and optimal cut-off values were estimated. 95% CI of AUC was calculated using the Mann-Whitney (MW) statistic approach, which has been suggested to be superior to other approaches for a small sample size. In a previous study, the MW statistic approach showed coverage probabilities close to the parametric approach of 95% CIs in small sample size [29]. SOFA scores, presepsin, IFN-λ3, and age were explored for predicting disease severity according to the COVID-19 severity index. AUC was interpreted as follows: 0.5 ≤ AUC < 0.6, poor; 0.6 ≤ AUC < 0.7, sufficient; 0.7 ≤ AUC < 0.8, good; 0.8 ≤ AUC < 0.9, very good; AUC ≥ 0.9, excellent [30]. AUC differences and *p* values were used to compare the discriminative ability of different models. In addition, the indicators were analyzed to find the optimal cut-off for predicting adverse clinical outcomes (in-hospital mortality, ICU admission, ventilator use, and kidney replacement therapy [KRT]). Next, the Kaplan-Meier survival analysis was performed, and using the optimal cut-off values, the hazard ratios (HR, 95% CI) of the SOFA score, presepsin, IFN-λ3, and age were obtained to compare the relative risk for predicting in-hospital mortality. The sample size for the Kaplan-Meier survival analysis was estimated based on a previous study [31]. The inputs were as follows: analysis time t = 2 months, accrual time α = 13 months, follow-up time b = 1 month, null survival probability, S_0_(t) = 0.013, 0.025, or 0.026, type I error rate (α) = 0.05, and the power (1 − β) = 0.8 [13]. The alternative survival probability was set to S_1_(t) = 0.236 based on the in-hospital mortality rate of this study. Using log-minus-log transformation that was suggested for improving accuracy in a small sample size, the estimated sample size was 11 to 15. Accordingly, the sample size of 55 was considered sufficient to perform the Kaplan-Meier survival analysis. Lastly, the in-hospital mortality rate based on the optimal cut-off values was compared according to the presepsin and IFN-λ3 by adjusting the age to over 65 years. For the statistical analyses, MedCalc Software (version 20.014, MedCalc Software, Ostend, Belgium) and R version 4.1.0 (The R Foundation for Statistical Computing, Vienna, Austria) were used. *p* < 0.05 was considered statistically significant.

## 3. Results

In the study population, sepsis or septic shock was observed in 34 patients (61.8%), with a proportion of 43.7% (*n* = 7) in non-critical disease patients and 69.2% (*n* = 27) in critical disease patients. Age, heart rate, respiratory rate, PaO_2_/FiO_2_, SOFA score, lymphocyte, blood urea nitrogen, creatinine, presepsin, IFN-λ3, C-reactive protein, prothrombin time, and activated partial thromboplastin time showed significant differences between non-critical and critical diseases (Table 1). Regarding the age, the proportion of elderly patients showed no significant difference between non-critical and critical diseases (*p* = 0.115).

Presepsin levels differed significantly between non-critical and critical diseases: 282 pg/mL (175–579 pg/mL) in the non-critical disease group vs. 558 pg/mL (402–1232 pg/mL) in the critical disease group (*p* = 0.008). IFN-λ3 levels also differed significantly between the two groups: 3.0 pg/mL (3.0–3.7 pg/mL) in the non-critical disease group vs. 6.9 pg/mL (3.0–13.8 pg/mL) in the critical disease group (*p* = 0.049) (Table 1 and Figure 1). In the ROC curve analysis, SOFA scores, presepsin, IFN-λ3, and age demonstrated sufficient to good performances for predicting disease severity (AUC = 0.67–0.73), and their performances were comparable to one another (Figure 2).

For predicting adverse clinical outcomes, SOFA scores and IFN-λ3 levels were comparable. SOFA scores demonstrated sufficient to very good performance (AUC = 0.68–0.88), and IFN-λ3 levels demonstrated sufficient to good performance (AUC = 0.66–0.74). On the contrary, presepsin levels demonstrated good performance (AUC = 0.74) only for in-hospital mortality, and age showed sufficient to very good performances (AUC = 0.66–0.83), except in cases of KRT. SOFA scores were superior to presepsin levels and age for predicting ventilator use (Figure 3). In the Kaplan-Meier analysis, HR (95% CI) for predicting in-hospital mortality was 3.6 (1.1–12.1) for SOFA scores, 6.7 (1.8–24.1) for presepsin and IFN-λ3, 2.8 (0.8–9.6) for age, 8.5 (6.8–24.6) for SOFA scores and presepsin + IFN-λ3, and 4.2 (0.9–20.6) for age and presepsin + IFN-λ3 (Figure 4).

Regarding mortality rates, the mortality rate was 23.6% (13/55) in the total patients and differed significantly according to biomarker levels. It was significantly higher when both presepsin and IFN-λ3 levels increased above optimal cut-offs than when either one or no biomarker level increased (81.8% vs. 9.1%, *p* < 0.001). The elderly patients accounted for 67.3% (37/55) of the total patients and 92.3% (12/13) of overall deaths. The mortality rate was 32.4% (12/37) in elderly patients. Similar to the total patients, the mortality rate was significantly higher when both presepsin and IFN-λ3 levels increased than when either one or no biomarker level increased (88.9% vs. 14.3%, *p* < 0.001) (Table 2).

## 4. Discussion and Conclusions

To the best of our knowledge, this is the first study that explored the clinical utility of two novel biomarkers, presepsin and IFN-λ3, compared with clinical assessments in hospitalized COVID-19 patients. In the present study, among the 55 COVID-19 patients, 16 (39.1%) patients belonged to the non-critical disease group and 39 (70.9%) patients belonged to the critical disease group [11]. Sepsis or septic shock was observed in 34 (61.8%) patients (*n* = 7, non-critical disease; *n* = 27, critical disease) (Table 1) [6].

In COVID-19, monocytes are the principal players in orchestrating the host’s immune response; thus, it can be expected that presepsin, which is secreted by monocytes, may have clinical utility in COVID-19 patients [5]. IFN-λ3 is a part of type III IFNs (IFN-λ1 [IL-29], -λ2 [IL-28A], -λ3 [IL-28B], and -λ4), which are structurally related to IL-10 family cytokines. In particular, type I and III IFNs may play crucial roles in innate immune responses [20,21,22]. In the present study, both presepsin and IFN-λ3 levels showed significant differences according to COVID-19 severity (Figure 1). The presepsin level was nearly 2-fold higher (mean difference of 276 pg/mL) and the IFN-λ3 level was nearly 2.5-fold higher (mean difference of 3.9 pg/mL) in the critical disease group than in the non-critical disease group. Our findings are in line with previous findings on each biomarker [13,16,17,18,23,24]. Of note, both presepsin and IFN-λ3 were comparable to SOFA scores and age for predicting disease severity (Figure 2); this finding implies that both biomarkers can be objective and reliable indicators in assessing the clinical status of COVID-19 patients.

Both SOFA scores and age demonstrated sufficient to very good performances (AUC = 0.68–0.88 and AUC = 0.66–0.83, respectively) for predicting all adverse clinical outcomes, with the exception of KRT (Figure 3). Among the two biomarkers, the IFN-λ3 level was comparable to SOFA scores for predicting adverse clinical outcomes (AUC = 0.66–0.74), although the presepsin level demonstrated good performance (AUC = 0.74) only for in-hospital mortality. Our findings support previous studies that demonstrated the clinical utility of presepsin for predicting disease severity and mortality (both in-hospital and 30-day) in critically ill patients [13,16,17,18,19]. Considering that the labeling and interpretation of AUC values may be arbitrary and the clinical usefulness of the discriminating ability of AUC values should be assessed using other tools including net benefit, further studies are warranted [32].

It was also noteworthy that, in the survival analysis, the combination of presepsin and IFN-λ3 showed higher HR (6.7; CI, 1.8–24.1) than SOFA scores (3.6, 1.1–12.1) or age (2.8, 0.8–9.6). This suggests that the combination of these biomarkers could be useful for predicting in-hospital mortality compared with clinical assessments such as SOFA scores. Due to the fact that biomarkers can be measured early at admission, they have advantages over clinical assessments and scoring systems that are difficult to calculate and take some time to evaluate [33,34]. Our data also highlighted that these biomarkers showed greater risk of mortality when added to clinical assessments or age (HR [95% CI], SOFA scores and presepsin and IFN-λ3, 8.5 [6.8–24.6]; age and presepsin and IFN-λ3, 4.2 [0.9–20.6]) (Figure 4).

We further compared in-hospital mortality rates according to biomarker levels (Table 2). Since elderly patients account for a significant proportion of deaths, it is important to stratify risk of in-hospital mortality rates in that age group [2,3]. Considering the high rate of comorbidities in elderly patients, there is a considerable need to investigate the clinical risk and aggressiveness of COVID-19 in these patients [35]. Our data also showed a significantly higher median age in the critical group than in the non-critical group and a higher mortality rate in the elderly patients than in the total patients (32.4% vs. 23.6%); among the 13 non-survivors, except for one patient, the other 12 patients were all elderly patients. In our study population, when both presepsin and IFN-λ3 levels increased above the optimal cut-offs (presepsin > 493 pg/mL and IFN-λ3 > 5.0 pg/mL), the mortality rate was approximately 90% in the elderly patients. Conversely, it was 14.3% in those elderly patients when the level of either one or both biomarkers was below cut-offs. This remarkable difference of mortality rate suggests that the combination of presepsin and IFN-λ3 can be used to stratify the risk of in-hospital mortality even in the same age group, and especially in elderly patients. Previously, the VACO index was developed to stratify the risk of 30-day mortality in same-age groups [8]. One study demonstrated that the combination of procalcitonin, presepsin, and the VACO index predicted 30-day mortality better than the VACO index alone in COVID-19 patients [13]; that finding also suggested that combinations of biomarkers with clinical assessments could be useful to stratify mortality risk in the same age group.

The present study has several limitations. First, it was a small-sized, single center study. We focused on identifying the distributions of each indicator according to the severity of COVID-19 and finding the optimal cut-off for predicting clinical outcomes. Information such as minimum effect size and difference are required to calculate the sample size as well as alpha and power. However, there is insufficient information on the differences between presepsin and IFN-λ3 in COVID-19 cases. Therefore, we used the ROC and Kaplan Meier methods, which are non-parametric survival analyses. Although there was no significant difference in the elderly groups between critical and non-critical diseases, the sample size was small. Therefore, it is necessary to conduct similar studies featuring larger populations to obtain more persuasive and reliable conclusions. Since this was an observational study, confounding variables could be eliminated via various methods such as propensity score matching; however, the sample size was relatively small, so this test was not performed. Therefore, there are limitations in that confounding controls need to be considered. Second, our study was conducted only on adult COVID-19 patients and had a skewed distribution towards the elderly (*p* = 0.003); accordingly, our data may not be representative of the entire age group. Due to the fact that COVID-19 could cause serious results in children, including multisystem inflammatory syndrome, it is also important to predict disease severity and adverse clinical outcomes in pediatric patients [4]. Third, although the clinical assessment and biomarker testing were conducted simultaneously at time of admission, the heterogeneous disease course of COVID-19 might have affected our data. Fourth, we focused on biomarker levels upon admission without monitoring their sequential changes, which might provide better clinical utility, especially using IVD assays on an automated platform in clinical laboratories. The possibility of using such biomarkers in predicting and stratifying disease severity not only in COVID-19 but also in other diseases should be explored in future studies; furthermore, it would be interesting to evaluate the function of such markers as possible therapeutic targets.

In conclusion, presepsin and IFN-λ3 predicted disease severity and adverse clinical outcomes in hospitalized COVID-19 patients. Both biomarkers, whether assessed alone or added to clinical assessments such as SOFA scores and age, are useful to assess and manage COVID-19 patients, especially the elderly. Further studies are awaited that explore the clinical utility of these novel biomarkers in critically ill patients and that validate our findings.

## Figures and Tables

**Figure 1 diagnostics-13-02372-f001:**
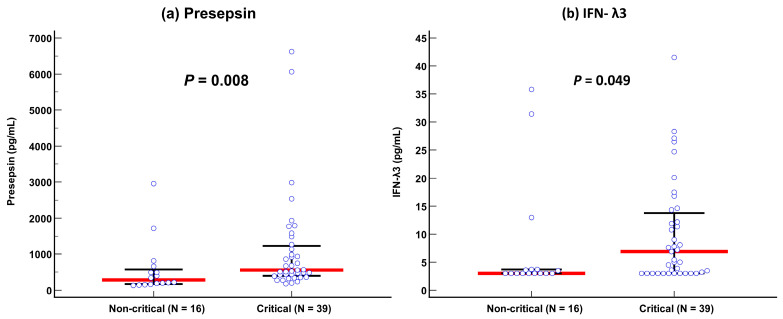
Comparison of presepsin and IFN-λ3 levels between non-critical and critical diseases according to the COVID-19 severity index. (**a**) Presepsin. (**b**) Interferon-λ3. Data are presented as median (interquartile range) using horizontal bars. Blue circles represent each individual, red lines represent medians, and black lines represent IQRs.

**Figure 2 diagnostics-13-02372-f002:**
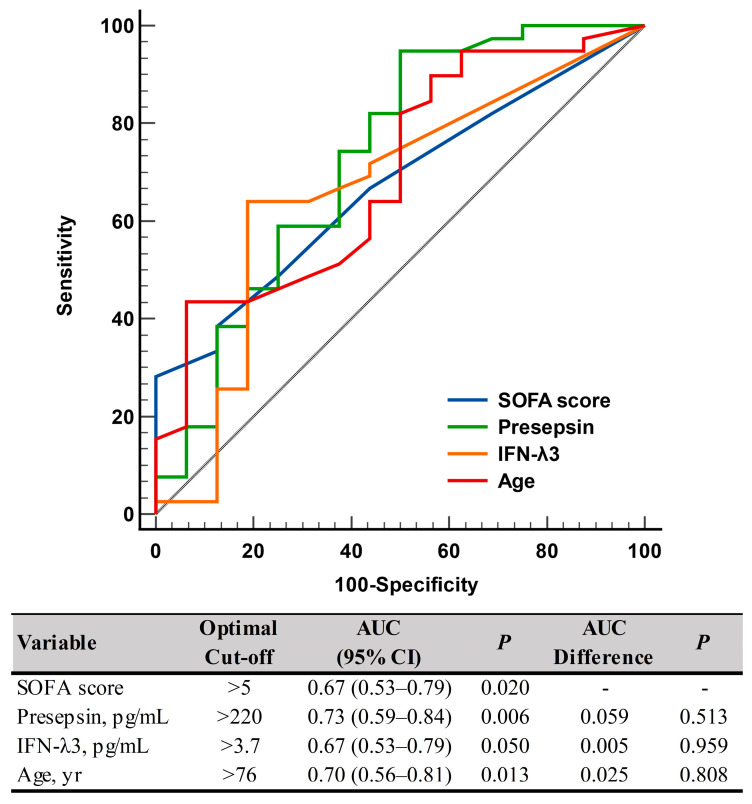
ROC curve analysis of the SOFA scores, presepsin, IFN-λ3, and age for predicting disease severity according to the COVID-19 severity index (non-critical vs. critical). Abbreviations: ROC, receiver operating characteristic; SOFA, sequential organ failure assessment; IFN-λ3, interferon-λ3; AUC, area under the curve; CI, confidence interval.

**Figure 3 diagnostics-13-02372-f003:**
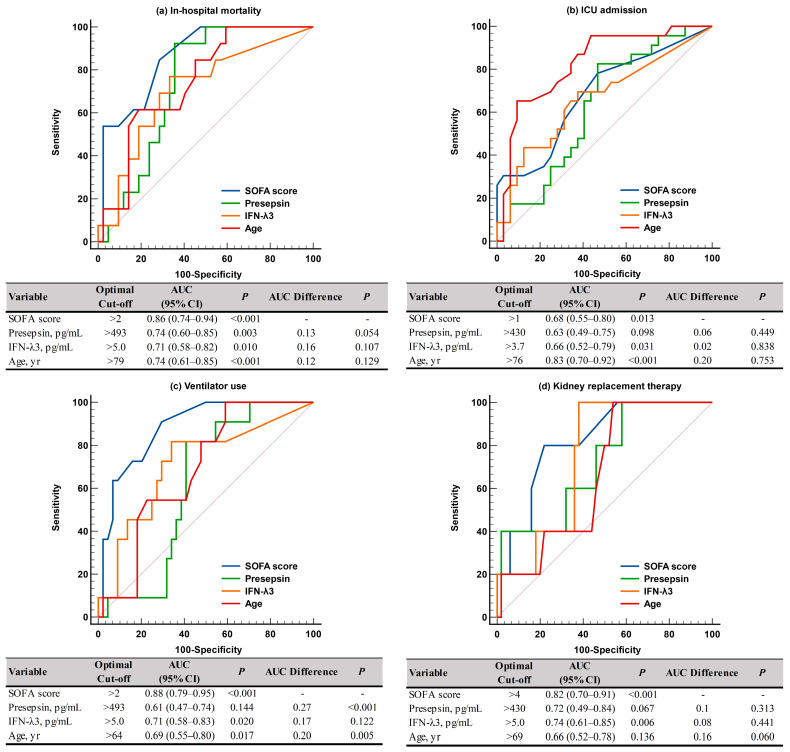
ROC curve analysis of the SOFA scores, presepsin, IFN-λ3, and age for predicting adverse clinical outcomes. (**a**) In-hospital mortality. (**b**) ICU admission. (**c**) Ventilator use. (**d**) Kidney replacement therapy. Abbreviations: see Figure 2; ICU, intensive care unit.

**Figure 4 diagnostics-13-02372-f004:**
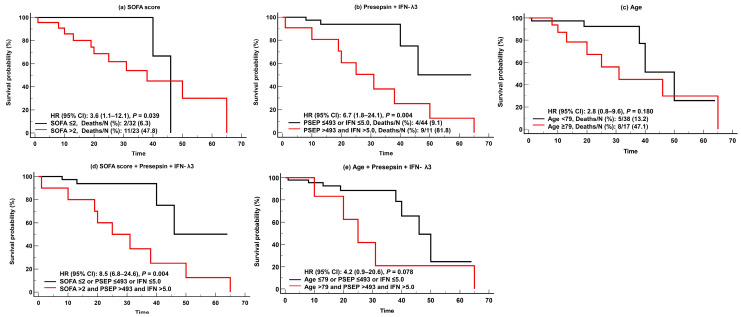
Kaplan-Meier survival analysis of the SOFA scores, presepsin, IFN-λ3, and age for predicting in-hospital mortality. (**a**) SOFA scores. (**b**) Presepsin + IFN-λ3. (**c**) Age. (**d**) SOFA scores and Presepsin and IFN-λ3. (**e**) Age and Presepsin and IFN-λ3. Abbreviations: PESP, presepsin; IFN, interferon-λ3; SOFA, sequential organ failure assessment; HR, hazard ratio; CI, confidence interval.

**Table 1 diagnostics-13-02372-t001:** Demographic, clinical, and laboratory findings of the study population.

Variable	Total (*n* = 55)	* Non-Critical (*n* = 16)	Critical (*n* = 39)	*p*
Age (years)	72 (61–80)	65 (47–75)	74 (65–82)	0.023
≥65, *n* (%)	37 (67.3)	8 (50.0)	29 (74.4)	0.115
<65, *n* (%)	18 (32.7)	8 (50.0)	10 (25.6)	0.115
Male, *n* (%)	30 (54.5)	7 (43.7)	23 (59.0)	0.377
Comorbidities, *n* (%)				
At least 2 comorbidities	28 (50.9)	5 (31.2)	23 (59.0)	0.079
Diabetes mellitus	20 (36.4)	5 (31.2)	15 (38.5)	0.761
Dementia	13 (23.6)	3 (18.8)	10 (25.6)	0.734
Heart failure	7 (12.7)	0 (0.0)	7 (17.9)	0.093
Malignancy	7 (12.7)	4 (25.0)	3 (7.7)	0.175
Liver disease	5 (9.1)	2 (12.5)	3 (7.7)	0.622
Chronic obstructive pulmonary disease	4 (7.3)	0 (0.0)	4 (10.3)	0.311
Chronic kidney disease	3 (5.5)	0 (0.0)	3 (7.7)	0.548
Cerebrovascular accident	2 (3.6)	0 (0.0)	2 (5.1)	1.000
Myocardial infarction	1 (1.8)	0 (0.0)	1 (2.6)	1.000
Chief complaints, *n* (%)				
Fever (>37.5 °C)	23 (41.8)	8 (50.0)	15 (38.5)	0.550
Chill	10 (18.2)	1 (6.2)	9 (23.1)	0.250
^†^ Respiratory symptoms	9 (16.4)	5 (31.2)	4 (10.3)	0.103
Dyspnea	7 (12.7)	0 (0.0)	7 (17.9)	0.093
Headache	4 (7.3)	1 (6.2)	3 (7.7)	1.000
^†^ Gastrointestinal symptoms	2 (3.6)	1 (6.2)	1 (2.6)	0.501
ICU stay (day)	0 (0–14)	0 (0–0)	2 (0–16)	0.024
Hospital stay (day)	17 (11–31)	13 (8–25)	19 (12–37)	0.045
COVID-19 diagnosis to admission (day)	0 (0–5)	0 (0–1)	0 (0–6)	0.244
COVID-19 diagnosis to blood sampling (day)	1 (0–6)	1 (0–3)	2 (0–7)	0.177
Vital signs				
Mean arterial pressure (mm Hg)	90 (84–98)	87 (82–93)	91 (84–100)	0.201
Heart rate (frequency/min)	86 (70–105)	75 (68–86)	93 (70–111)	0.041
Respiratory rate (frequency/min)	20 (20–22)	20 (20–20)	21 (20–25)	0.005
Body temperature (°C)	36.7 (36.5–37.6)	36.5 (36.4–37.6)	36.8 (36.5–37.6)	0.190
PaO_2_/FiO_2_ (mm Hg)	377 (309–445)	460 (374–471)	358 (285–436)	0.001
Clinical outcomes				
ICU admission, *n* (%)	23 (41.8)	3 (18.8)	20 (51.3)	0.036
In-hospital mortality, *n* (%)	13 (23.6)	0 (0.0)	13 (33.3)	0.011
Ventilator use, *n* (%)	11 (20.0)	0 (0.0)	11 (28.2)	0.023
Kidney replacement therapy, *n* (%)	5 (9.1)	0 (0.0)	5 (12.8)	0.306
SOFA score	2 (1–5)	1 (0–3)	2 (1–7)	0.045
Central nervous system	0 (0–1)	0 (0–0)	0 (0–1)	0.039
Renal	0 (0–0)	0 (0–0)	0 (0–1)	0.019
Liver	0 (0–0)	0 (0–0)	0 (0–0)	0.809
Circulatory	0 (0–0)	0 (0–0)	0 (0–1)	0.189
Coagulation	0 (0–1)	0 (0–1)	0 (0–1)	0.625
Respiratory	1 (0–1)	0 (0–1)	1 (0–2)	0.039
Sepsis or septic shock, *n* (%)	34 (61.8)	7 (43.7)	27 (69.2)	0.126
Laboratory data				
White blood cell (×10^9^/L)	6.4 (4.6–11.3)	6.3 (4.3–9.5)	6.6 (4.7–13.9)	0.535
Neutrophil (×10^9^/L)	5.2 (3.3–9.9)	4.1 (2.4–7.9)	5.5 (3.4–13.2)	0.258
Lymphocyte (×10^9^/L)	0.9 (0.6–1.4)	1.3 (0.9–1.7)	0.8 (0.5–1.2)	0.008
Monocyte (×10^9^/L)	0.4 (0.3–0.7)	0.4 (0.3–0.6)	0.4 (0.3–0.8)	0.767
Red blood cell (×10^12^/L)	3.9 (3.5–4.6)	3.9 (3.6–4.3)	3.9 (3.4–4.6)	0.978
Platelet (×10^9^/L)	186 (143–261)	179 (149–271)	188 (136–266)	0.846
Total bilirubin (umol/L)	0.6 (0.4–0.9)	0.6 (0.4–0.8)	0.6 (0.4–0.9)	0.493
Blood urea nitrogen (mmol/L)	17.5 (13.7–27.6)	12.3 (8.5–17.4)	18.7 (15.2–35.5)	0.002
Creatinine (μmol/L)	0.8 (0.6–1.0)	0.7 (0.6–0.8)	0.8 (0.7–1.3)	0.031
C-reactive protein (mg/L)	6.2 (2.6–11.7)	2.8 (0.2–3.5)	8.0 (3.5–12.3)	0.004
Presepsin (pg/mL)	493 (330–963)	282 (175–579)	558 (402–1232)	0.008
Interferon-λ3 (pg/mL)	3.9 (3.0–12.1)	3.0 (3.0–3.7)	6.9 (3.0–13.8)	0.049
Lactate (mmol/L)	2.0 (1.8–2.6)	2.0 (1.4–2.0)	2.1 (1.8–2.9)	0.056
Prothrombin time (s)	13.0 (12.3–14.6)	12.4 (11.8–13.0)	13.6 (12.5–15.3)	0.001
Activated partial thromboplastin time (s)	39.7 (33.4–44.1)	33.7 (29.5–39.7)	40.6 (35.8–45.6)	0.005
D-dimer (ug/mL)	1.6 (0.7–3.6)	1.6 (0.6–3.5)	1.6 (0.9–3.6)	0.434

Data are expressed as a number (percentage) or median (interquartile range). * The total population was divided into two groups according to the COVID-19 severity index, and non-critical disease includes low (*n* = 3), moderate (*n* = 11), and high (*n* = 2) in the COVID-19 severity index. ^†^ Respiratory symptoms include cough, sputum, rhinorrhea, and sore throat; gastrointestinal symptoms include abdominal pain and diarrhea. Abbreviations: SOFA, sequential organ failure assessment; ICU, intensive care unit.

**Table 2 diagnostics-13-02372-t002:** Comparison of in-hospital mortality rate according to the presepsin and interferon-λ3 levels using optimal cut-off values.

	PSEP ≤ 493 pg/mL or IFN ≤ 5.0 pg/mL	PSEP > 493 pg/mL and IFN > 5.0 pg/mL	*p*
Overall (*n* = 55)			
Survivor (*n* = 42)	40 (95.2)	2 (4.8)	-
Non-survivor (*n* = 13)	4 (30.8)	9 (69.2)	-
Mortality rate, proportion (%)	4/44 (9.1)	9/11 (81.8)	<0.001
Age ≥ 65 years (*n* = 37)			
Survivor (*n* = 25)	24 (96.0)	1 (4.0)	-
Non-survivor (*n* = 12)	4 (33.3)	8 (66.7)	-
Mortality rate, proportion (%)	4/28 (14.3)	8/9 (88.9)	<0.001

Data are expressed as a number (percentage). The mortality rates were 23.6% (13/55) in the total patients and 32.4% (12/37) in elderly patients (≥65 years). Abbreviations: PESP, presepsin; IFN, interferon-λ3.

## Data Availability

The data presented in this study are available from the corresponding author on reasonable request.

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
