# Peer review of "Utility of Presepsin and Interferon-λ3 for Predicting Disease Severity and Clinical Outcomes in COVID-19 Patients"

_diagnostics, 2023, doi:10.3390/diagnostics13142372_

Round 1

Reviewer 1 Report

Lee et al. presented the manuscript titled: "Utility of presepsin and interferon-λ3 for predicting disease severity and clinical outcomes in COVID-19 patients".

The manuscript is clear, well written and topic is interesting. They represent the data about new biomarkers, presepsin and INF-λ3, for prediction of disease severity.

Although, the cohort includes only 55 patients, the results are interesting and gives a potential to perform the same analysis for the larger cohort of patients. 

Both mentioned biomarkers, alone or added to other clinical assessments such as SOFA score and age, could be useful to assess and manage COVID-19 patients, especially older population. 

In conclusion, the topic is very interesting and important for further applications and manuscript can be accepted in present form.

Author Response

Thank you very much for your time and effort for reviewing our manuscript.

Reviewer 2 Report

The study investigated the utility of presepsin and interferon-λ3 as biomarkers for predicting disease severity and clinical outcomes in hospitalized COVID-19 patients. The results showed that both biomarkers could predict disease severity and clinical outcomes, with presepsin being particularly useful in predicting in-hospital mortality. The combination of presepsin and IFN-λ3 showed a higher mortality risk than SOFA score or age, which suggests that these biomarkers could be valuable additions to clinical assessments for managing COVID-19 patients, especially in the elderly. However, there are a few minor revisions that need to be made before the manuscript can be published.

1. For a small-scale study, 55 patients is already a considerable sample size. However, in scientific research, larger-scale studies are typically required to obtain more persuasive and reliable conclusions. The authors are advised to reconsider whether 55 samples are optimal for their analysis.

2. Line numbers were not visible in the PDF version of the manuscript, which makes it difficult for readers and reviewers to refer to specific parts of the article. The authors are advised to include line numbers in future versions of the manuscript.

3. This study is relatively simple, and from a statistical perspective, the experimental design is not particularly robust, and the analysis is limited. As mentioned in the manuscript, there are many limitations to this study. However, given the limitations of the authors' abilities and the sample size, the authors are encouraged to consider additional analyses from different perspectives to minimize the limitations of this manuscript.

4. The authors are encouraged to consider whether the limitations mentioned in the manuscript can be minimized. For example, can more standardized methods be used to collect data on the time from symptom onset to biomarker testing to reduce data bias? Can the continuous changes in biomarker levels be monitored in the study to assess their practical value in managing COVID-19 patients?

5. To facilitate understanding and readability of the manuscript, it is recommended that the statistical methods used in the study be described in more detail for readers.

6. The advantages and limitations of the selected testing methods can be briefly described, and the rationale for choosing these methods can be explained.

7. Please check the formatting at the end of Table 1.

8. Please improve the clarity of Figures 3 and 4.

9."Added on the clinical assessment, 'on' should be changed to 'to'."

10."In the sentence 'older account for a strikingly higher proportion of COVID19 deaths (81.0 - 86.2%)', 'COVID19' should be changed to 'COVID-19', which is consistent with the terminology used in the abstract."

11."In the sentence 'COVID-19 patients show heterogenous disease course ranging from no symptom to critical illness', 'course' should be changed to 'courses'."

12."In the phrase 'for small sample size', 'small' should be preceded by 'a'."

13."In the phrase 'analysis time t = 2 month', 'month' should be changed to 'months'."

14."In the phrase 'followup time b = 1 month', 'followup' should be changed to 'follow-up'."

15."In the phrase 'with the proportion of 43.7% (n = 7)', 'the' should be changed to 'a'."

16."In the sentence 'their performances were comparable one another (Figure 2)', 'one another' should be changed to 'to one another' (Figure 2)."

17."In the phrase 'and age for predicting inhospital mortality', 'inhospital' should be changed to 'in-hospital'."

18."In the sentence 'Because biomarkers can be measured easily on early time of admission,'on early time' should be changed to 'early on'."

19."In the sentence 'it has advantages than clinical assessments', 'than' should be changed to 'over'."

20."In the sentence 'the level of either one or both biomarkers were below cut-offs', 'were' should be changed to 'was'."

There are some grammatical mistakes and wordy expressions in the manuscript, which should be carefully corrected to be succinct.

Author Response

  1. For a small-scale study, 55 patients is already a considerable sample size. However, in scientific research, larger-scale studies are typically required to obtain more persuasive and reliable conclusions. The authors are advised to reconsider whether 55 samples are optimal for their analysis.

Thank you for your comment. According to your comment, we rechecked the appropriateness of sample number and we added one more co-author (Lee S) who is the expert statistician. We also modified the following sentence to reflect your opinion.

Although the size of study population was sufficient for statistical analysis, further studies with larger sample size would be necessary to obtain more persuasive and reliable conclusions. (page 13)

  1. Line numbers were not visible in the PDF version of the manuscript, which makes it difficult for readers and reviewers to refer to specific parts of the article. The authors are advised to include line numbers in future versions of the manuscript.

Thank you for your comment. Line numbers were included in our original manuscript that was submitted. We think that the publisher used its own manuscript format for the review process, and the line numbers were eliminated during that process. For the revision, we also had to use the publisher-provided manuscript format; so, please understand that we could not include the line numbers. Instead, we highlighted the changed portions using red-colored text.

  1. This study is relatively simple, and from a statistical perspective, the experimental design is not particularly robust, and the analysis is limited. As mentioned in the manuscript, there are many limitations to this study. However, given the limitations of the authors' abilities and the sample size, the authors are encouraged to consider additional analyses from different perspectives to minimize the limitations of this manuscript.

Thank you for your comment. According to your valuable comment, we modified Table 1 and included 3 more variables (at least 2 comorbidities, ICU stay, and hospital stay) in Table 1.

  1. The authors are encouraged to consider whether the limitations mentioned in the manuscript can be minimized. For example, can more standardized methods be used to collect data on the time from symptom onset to biomarker testing to reduce data bias? Can the continuous changes in biomarker levels be monitored in the study to assess their practical value in managing COVID-19 patients?

Thank you for your comment. According to your valuable comment, we modified the following sentences in Methods and Discussion sections.

The study population had no specific limitations on care at enrollment, and the biomarker testing was conducted at enrollment (IQR, 0-1 day). (page 2)

The SOFA score and CSI were assessed at enrollment together with biomarker testing. (page 3)

Third, although the clinical assessment and biomarker testing were conducted simultaneously at early time of admission, the heterogeneous disease course of COVID-19 might have affected our data. (page 13)

  1. To facilitate understanding and readability of the manuscript, it is recommended that the statistical methods used in the study be described in more detail for readers.

According to your comment, we modified the statistical analysis section extensively.

Data are presented as number (percentage) or median (IQR). The Mann–Whitney U test and Fisher’s exact test were used to compare the demographic variables, clinical out-comes and laboratory data between non-critical (n = 16) and critical (n = 39) diseases. The distribution of presepsin and IFN- λ3 were presented for assessing the normality and detecting outliers using the Shapiro–Wilk test and the Grubb’s test. Receiver operating characteristic (ROC) curves were analyzed for predicting disease severity and clinical out-comes, and the area under the curve (AUC) and optimal cut-off values were estimated. 95% CI of AUC was calculated using the Mann-Whitney statistic approach, which was suggested to be superior to others for a small sample size [29]. SOFA score, presepsin, IFN- λ3, and age were explored for predicting disease severity according to the COVID-19 severity index. AUC was interpreted as follows: 0.5 ≤ AUC < 0.6, poor; 0.6 ≤ AUC < 0.7, sufficient; 0.7 ≤ AUC < 0.8, good; 0.8 ≤ AUC < 0.9, very good; AUC ≥ 0.9, excellent [30]. AUC differences and P values were used to compare the discriminative ability of different models. In addition, the indicators were analyzed to find the optimal cut-off for predicting adverse clinical outcomes (in-hospital mortality, ICU admission, ventilator use, and kidney replacement therapy [KRT]). Next, Kaplan-Meier survival analysis was performed, and using optimal cut-off values, hazard ratios (HR, 95% CI) of the SOFA score, presepsin, IFN- λ3, and age were obtained to compare the relative risk for predicting in-hospital mortality. The sample size for the Kaplan-Meier survival analysis was estimated based on the previous study [31]. The inputs were as follows: analysis time t = 2 months, accrual time α = 13 months, follow-up time b = 1 month, null survival probability, S0(t) = 0.013, 0.025, or 0.026, type â…  error rate (α) = 0.05, and the power (1 – β) = 0.8 [13]. The alternative survival probability was set to set to S1(t) = 0.236 based on in-hospital mortality of this study. Using log-minus-log transformation suggested for improving accuracy in small sample size, the estimated sample size was 11 to 15. Accordingly, the sample size of 55 was considered sufficient to perform the Kaplan-Meier survival analysis. Lastly, the in-hospital mortality based on the optimal cut-off values are compared according to the presepsin and IFN- λ3 by adjusting age over 65 years. For the statistical analyses, MedCalc Software (version 20.014, MedCalc Software, Ostend, Belgium) and R version 4.1.0 (The R Foundation for Statistical Computing, Vienna, Austria) were used. P < 0.05 was considered statistically significant.

  1. The advantages and limitations of the selected testing methods can be briefly described, and the rationale for choosing these methods can be explained.

According to your comment, we added the following sentences in Discussion section. We added a new reference.

Considering that the labeling and interpretation of AUC values may be arbitrary and clinical usefulness of discriminating ability of AUC values should be assessed using other tools including net benefit, further studies would be warranted [32]. (page 12)

We focused on identifying the distributions of each indicator according to the severity of COVID-19 and finding the optimal cut-off for predicting clinical outcomes. Information such as minimum effect size and difference are required to calculate the sample size as well alpha, and power. However, there is insufficient information on the differences of presepsin and IFN-λ3 in COVID-19 cases. Therefore, we used ROC and Kaplan Meier method which is a non-parametric survival analysis. (page 13)

  1. de Hond, A.A.H.; Steyerberg, E. W.; van Calster, B. Interpreting area under the receiver operating characteristic curve. Lancet. Digit. Health. 2022, 4, e853-e855.

  1. Please check the formatting at the end of Table 1.

According to your comment, we checked the format of Table 1.

  1. Please improve the clarity of Figures 3 and 4.

According to your comment, we improved the quality of all the figures (improved dpi). Because we could not merge the new figures into the publisher-provided manuscript format, we uploaded the figures separately.

9."Added on the clinical assessment, 'on' should be changed to 'to'."

According to your comment, we changed it.

10."In the sentence 'older account for a strikingly higher proportion of COVID19 deaths (81.0 - 86.2%)', 'COVID19' should be changed to 'COVID-19', which is consistent with the terminology used in the abstract."

According to your comment, we changed it.

11."In the sentence 'COVID-19 patients show heterogenous disease course ranging from no symptom to critical illness', 'course' should be changed to 'courses'."

According to your comment, we changed it.

12."In the phrase 'for small sample size', 'small' should be preceded by 'a'."

According to your comment, we changed it.

13."In the phrase 'analysis time t = 2 month', 'month' should be changed to 'months'."

According to your comment, we changed it.

14."In the phrase 'followup time b = 1 month', 'followup' should be changed to 'follow-up'."

According to your comment, we changed it.

15."In the phrase 'with the proportion of 43.7% (n = 7)', 'the' should be changed to 'a'."

According to your comment, we changed it.

16."In the sentence 'their performances were comparable one another (Figure 2)', 'one another' should be changed to 'to one another' (Figure 2)."

According to your comment, we changed it.

17."In the phrase 'and age for predicting inhospital mortality', 'inhospital' should be changed to 'in-hospital'."

According to your comment, we changed it.

18."In the sentence 'Because biomarkers can be measured easily on early time of admission,'on early time' should be changed to 'early on'."

According to your comment, we changed it.

19."In the sentence 'it has advantages than clinical assessments', 'than' should be changed to 'over'."

According to your comment, we changed it.

20."In the sentence 'the level of either one or both biomarkers were below cut-offs', 'were' should be changed to 'was'."

According to your comment, we changed it.

Reviewer 3 Report

This article focuses on the possibility of using biomarkers such as presepin and lìinterferon lambda 3 to predict disease severity and clinical outcomes in coronavirus patients.

The authors assessed how, globally, elderly patients aged 65 years and older represent the percentage of patients with the highest mortality rate. In relation to clinical evaluations in these patients, they pointed out that in addition to the SOFA score, markers such as presepin and d interferon lamba 3 are also useful in predicting disease severity.

These include in particular presepin, released into the circulation by pro-inflammatory signals during infection, whose diagnostic and prognostic utility was evaluated not only in sespi but also in COVID 19.

The authors therefore focused on using these 2 biomarkers to better stratify clinical severity and mortality risk in patients with COVID 19.

Thus, in the analysed population, it was seen that presepin levels differed significantly between the group of patients with severe disease compared to the group of patients with severe disease.

Despite this very interesting and well-done article, there are some shortcomings, such as the possibility of using such biomarkers not only to predict and stratify disease severity in COVID 19, but also in other diseases, furthermore, it would be interesting to evaluate the function of such markers as possible therapeutic targets.

The manuscript needs a language check, but it is lacking in several points that would add value to the entire manuscript.

1.     Regarding the topics covered, it would be interesting to take a cue from this article 10.1159/000509434As it could give new insights to the article and enrich it with potential. 

Minor revisions.

Author Response

Despite this very interesting and well-done article, there are some shortcomings, such as the possibility of using such biomarkers not only to predict and stratify disease severity in COVID 19, but also in other diseases, furthermore, it would be interesting to evaluate the function of such markers as possible therapeutic targets.

Thank you for your comment. According to your comment, we added the following sentences in the Discussion section.

The possibility of using such biomarkers is predicting and stratifying disease severity not only in COVID-19 but also in other diseases; furthermore, it would be interesting to evaluate the function of such markers as possible therapeutic targets. (page 13)

The manuscript needs a language check, but it is lacking in several points that would add value to the entire manuscript.

Thank you for your comment. According to your comment, we did the language check again for the whole manuscript.

  1. Regarding the topics covered, it would be interesting to take a cue from this article 10.1159/000509434. As it could give new insights to the article and enrich it with potential.

Thank you for your comment. According to your comment, we added the following sentence in the Discussion section, and we also added the reference.

Considering the high rate of comorbidities in elderly patients, there is a considerable need to investigate the clinical risk and aggressiveness of COVID-19 in these patients [35].

  1. Di Lorenzo, G.; Buonerba, L.; Ingenito, C.; Crocetto, F.; Buonerba, C.; Libroia, A.; et al. Clinical characteristics of metastatic prostate cancer patients infected with COVID-19 in south Italy. Oncology. 2020, DOI: 10.1159/000509434

Round 2

Reviewer 3 Report

Authors answered all comments and suggestions.

Minor editing.

Author Response

Thank you very much for your review.

We did the language check again and improved the manuscript quality.